# Innovative Regional Services and Heterogeneous Communication Channels: Results from the Nationwide German egePan Project for Pandemic Management

**DOI:** 10.3390/healthcare12212192

**Published:** 2024-11-04

**Authors:** Simon Kugai, Benjamin Aretz, Yelda Krumpholtz, Manuela Schmidt, Daniela Süssle, Linda Steyer, Adrienne Henkel, Katrin Bender, Felix Girrbach, Sebastian Stehr, Katrin Balzer, Birgitta Weltermann

**Affiliations:** 1Institute of General Practice and Family Medicine, Medical Faculty, University of Bonn, Venusberg-Campus 1, 53127 Bonn, Germany; 2Nursing Research Unit, Institute for Social Medicine and Epidemiology, University of Lübeck, Ratzeburger Allee 160, 23562 Lübeck, Germany; 3Department of Anaesthesiology and Intensive Care, University of Leipzig Medical Center, Liebigstrasse 20, 04103 Leipzig, Germany; 4Anesthesiology and Operative Intensive Care, Faculty of Medicine, University of Augsburg, 86156 Augsburg, Germany

**Keywords:** pandemic preparedness, COVID-19, ambulatory care, hospitals, emergency medical services, nursing services, regional health planning, telemedicine

## Abstract

**Background**: In the COVID-19 pandemic, novel regional services and communication channels emerged across all sectors of the German healthcare system. To contribute to pandemic preparedness, this study aims to describe newly established services in response to the COVID-19 pandemic from a stakeholder perspective and to examine the interprofessional communication channels, applying a nationwide cross-sectional approach. **Methods**: A nationwide sample of German healthcare stakeholders comprising general practitioners, associations of statutory health insurance physicians, hospital medical directors, local health departments, rescue coordination centres, medical directors of emergency services, outpatient nursing services, nursing homes, community care access centres, and hospital nursing managers was surveyed. A web-based questionnaire asked for their level of participation in newly implemented regional COVID-19 services and communication channels. Stakeholders’ level of recommendation was measured using the Net Promotor Score (NPS), a metric that assesses their satisfaction towards the services surveyed. **Results**: In total, 1312 healthcare stakeholders participated in the survey. Diagnostic centres (23.0–90.9%), COVID-19 wards in hospitals (40.5–92.1%), emergency medical vehicles designated solely for COVID-19 patients (16.5–68.4%), and crisis intervention teams (11.6–30.6%) exhibited the highest rates of engagement. The services receiving the highest recommendation for future use were COVID-19 focus practices (NPS: 33.4–43.7), COVID-19 wards in hospitals (NPS: 47.6–84.4), transportation of COVID-19 patients exclusively by predefined professional groups (NPS: 12.5–36.4), and newly implemented digitally supported nursing services (NPS: 58.3–100.0). Telephones emerged as the most frequently used communication channel (58.0–96.7%), while email was the primary digital channel (23.7–81.5%). **Conclusions**: During the COVID-19 pandemic, Germany experienced significant variation in the implementation of pandemic-related services across healthcare sectors, with stakeholders prioritising services built on existing healthcare structures. Developing a proactive digital infrastructure to connect healthcare professionals from different sectors is crucial for better future pandemic management.

## 1. Introduction

The COVID-19 pandemic required efficient strategies for healthcare provision. Various new regional services were implemented in all levels of care, from emergency and outpatient to inpatient services. In the outpatient sector, new diagnostic centres in non-medical settings and dedicated COVID-19 diagnostic practices [1,2], “Corona-Taxis” for patient transport [3], specialised COVID-19 practices [4], outpatient clinics [5], and COVID-19 vaccination centres [6] were established. In the inpatient sector, COVID-19 wards in hospitals [7,8,9], specialised COVID-19 hospitals [10], temporary treatment centres in non-medical facilities [11,12], and temporary wards in rehabilitation hospitals [13] were crucial for managing the influx of patients. In the rescue sector, COVID-19 rescue coordination centres [14], specialised emergency vehicles [15,16,17], tele-emergency doctors for remote medical consultations [18,19], and dedicated medical response teams were mobilised to ensure efficient care and transportation of COVID-19 patients [20,21]. In the nursing sector, digital appointment scheduling [22], telenursing [23], telemedicine/telehealth services [24,25,26], preventive teams ensuring a safe working environment [27,28], crisis intervention teams addressing mental health crises among healthcare professionals [29], a COVID-Team-Time-Out as a joint checklist review of hygiene standards [30], and various other emergency nursing services [31,32] were established to adapt to the challenges posed by the pandemic.

Another strategy for healthcare delivery was to shift towards non-contact healthcare methods [33] and remote communication solutions [34]. Telecommunications were crucial in reducing personal contact while maintaining quality healthcare [35,36]. The development of digital messenger applications further supported this transition [37]. However, the utilisation of digital communication varied significantly between countries, primarily influenced by their digital infrastructure [38]. Some highly developed nations like Norway and Australia embraced digital solutions and telemedicine extensively, while others, like Germany, showed a limited implementation in outpatient healthcare [35,36,39,40]. Despite being a highly developed country, Germany’s slow adoption of digital health stems from strict data protection laws, insufficient technical guidelines, sluggish industry IT solutions, and outdated, inflexible systems in many medical practices [41]. This discrepancy highlights the urgent need for developing digital and remote health services [33].

While individual healthcare services and communication channels have been extensively studied in the existing literature [6,11,15,28], there is a notable research gap in Germany on pandemic preparedness. Studies primarily focused on the usage patterns of these services, their benefits, and areas for enhancement. Research has examined patient health trajectories [5,7,8] and management [10,31,35], communication dynamics between healthcare professionals and patients [36,39,42], and public communication strategies [43,44]. However, the exploration of communication dynamics among healthcare professionals themselves is needed. Despite the recognised importance of effective communication for improving quality and safety in healthcare [45], existing studies such as those conducted by Sheehan et al. (2021) [46] found a lack of methods or measures specifically addressing communication between inpatient and outpatient physicians. However, the current post-pandemic period allows for the reflection, adjustment, and improvement in future pandemic preparedness [47].

While many COVID-19 healthcare services have been individually explored in the literature, their comparative analysis, particularly regarding their actual benefit, remains underexplored. To contribute to pandemic preparedness, this study aims to describe newly established services in response to the COVID-19 pandemic from a stakeholder perspective and to examine the interprofessional communication channels applying a nationwide cross-sectional approach.

## 2. Materials and Methods

The egePan Unimed research project, which was initiated by the nationwide network of university medicine (NUM), aimed to develop, test, and implement new services for pandemic management, particularly during the COVID-19 pandemic. The project involved identifying and evaluating new services across various sectors of the healthcare system in Germany. The Ethics Committee of the Medical Faculty of the University of Bonn raised no objections to conducting the study (5 February 2021, Reference number: 419/20). Only participants who gave informed consent via a web-based survey were included. The study was conducted in line with the Declaration of Helsinki. The survey focused on the organisation and structure of the German healthcare system, examining outpatient, inpatient, rescue and nursing sectors. Different professional groups within each sector received tailored surveys. Participants included general practitioners (GPs), associations of statutory health insurance physicians (ASHIPs), hospital medical directors, rescue coordination centres, medical directors of emergency services (MDESs), outpatient nursing services, nursing homes, community care access centres (CCACs), hospital nursing managers, and local health departments.

### 2.1. Sampling Strategy

The sampling methods varied depending on the professional group of the invitees. GPs, forming the largest group, were chosen through a two-step process. Initially, we bought a complete list of all working German GPs from ArztData AG, a professional company for address management in healthcare, and categorized it into quartiles based on the federal state and regional population density, with 40% of counties randomly selected. Within these county clusters, GPs were further stratified by practice type and employment status, with 30% selected from each stratum. Contact details of other professionals such as ASHIPs, local health departments, hospital medical directors, MDESs, and CCACs were obtained online by manual research, ensuring coverage across all cities and counties of Germany. Hospital medical directors were only included if their hospital had at least one department for internal medicine and surgery. They were selected from different regions of Germany, with three cities or counties chosen from each region for invitation. Additionally, rescue coordination centres were invited via a newsletter from the federal association of rescue coordination centres with nationwide coverage. The sampling strategy aimed to ensure representation across urban and rural areas and various professional sectors within the healthcare system. After excluding incorrect email addresses and participants who requested to be removed from the study, the final numbers of eligible participants included 9287 GPs, 18 ASHIPs, 1123 hospital medical directors, 372 local health departments, 237 MDESs, 61 rescue coordination centres, 1153 outpatient nursing services, 944 nursing homes, 382 CCACs and 322 hospital nursing managers.

### 2.2. Questionnaire Design

A team of healthcare professionals created the web-based questionnaire based on a comprehensive literature review [10,12,15,17,38,48,49,50]. Given the dynamic nature of pandemic management, current public reporting and the authors’ knowledge were also important sources for designing the questionnaire. The team of authors covered expertise in outpatient and inpatient medical care, nursing sciences, rescue services, social sciences, and public healthcare. The main criteria for development included gathering key information about sociodemographic and workplace characteristics while ensuring participant anonymity, identifying structures specifically established to combat the COVID-19 pandemic with timeliness being very important, and covering common communication channels and partners. During development, an interim version of the questionnaire, hosted by the platform unipark.com, was piloted by 55 experts from the areas of general and internal medicine, virology, microbiology, hygiene, anaesthesiology, public health, psychiatry, and nursing. The results of this pilot phase were incorporated into the final questionnaire, which was open from 17 March 2021 until 17 June 2021. The invitations were emailed, and a reminder was issued after four weeks. Data collection was anonymous as far as possible. However, anonymity could not be guaranteed for ASHIPs and rescue coordination centres due to their small group size at the federal state level, so the federal states were combined into regions. Bremen, Hamburg, Lower Saxony, and Schleswig-Holstein formed the north; Berlin, Brandenburg, Mecklenburg-Western Pomerania, Saxony, Saxony-Anhalt and Thuringia the east; Hesse, North-Rhine Westphalia, Rhineland Palatinate and Saarland the west; and Baden-Württemberg and Bavaria the south region. The questionnaire is publicly available on Mendeley Data [51].

### 2.3. Sociodemographic Characteristics

The participants’ sociodemographic characteristics were assessed in terms of gender, years in their current professional position, and region of residence. GPs were asked about how many years they had been licensed to treat patients insured through statutory health insurance funds and bill accordingly. All participants were queried to disclose whether they used the German COVID-19 tracing app (Corona-Warn-App), which has received multiple updates each month. Work-related details were documented for the groups of GPs, hospital medical directors, local health departments, rescue coordination centres, outpatient nursing services, nursing homes, CCACs, and hospital nursing managers. Each group reported the total number of full-time and part-time employees, with outpatient nursing services, nursing homes, CCACs, and hospital nursing managers detailing their nursing staff count. Moreover, hospital medical directors, local health departments, and rescue coordination centres provided information about the number of new hires responding to the COVID-19 pandemic.

### 2.4. Newly Implemented Regional COVID-19 Services and Communication Channels

Participants were surveyed regarding their engagement in new regional services to address the pandemic’s challenges across different healthcare sectors. The new services elicited in the questionnaire were researched in advance on the basis of scientific literature and media reports. GPs, ASHIPs, and local health departments were questioned about eight new outpatient services. Hospital medical directors and local health departments were queried about five new inpatient services. Hospital medical directors were additionally asked about diagnostic centres as outpatient services. Local health departments, MDESs, and rescue coordination centres were asked about seven new rescue services. Outpatient nursing services, nursing homes, CCACs, and hospital nursing managers were surveyed about nine new nursing services.

Possible answers for outpatient and inpatient structures were “regionally present, involved” (A), “regionally present, not involved” (B), and “not existent or known” (D). Local health departments also had options A, B, and D for rescue services. MDESs and rescue coordination centres could choose between “regionally present” (C) and D. Nursing services had only options C and D. Answering options were simplified from A and B to C, if it could be assumed that the knowledge of a change implied involvement. If participants selected A or C, a further question appeared to assess the extent to which the structure would be recommended for further pandemic response on a scale from 0 (“no recommendation”) to 10 (“absolute recommendation”).

In addition, the questionnaire explored the use of various communication channels. Each participant disclosed their preferred communication channels for engaging with patients and healthcare professionals. They were presented with various potential communication partners, with whom collaboration or certain overlaps could occur. In selecting their communication channels, participants had options including “on-site”, “telephone”, “fax”, “email”, “video call”, “messenger app”, “other”, and “not specified”, with the possibility of choosing multiple answers per communication partner.

### 2.5. Statistical Analysis

The first analysis step computed frequencies, percentages, mean values, and standard deviations for sociodemographic and workplace characteristics to describe the sample. Percentages were calculated for new regional services and communication channels. The comparison between participants and nonparticipants was performed by stratifying for federal states and gender. A Chi-squared test examined the gender distribution, and an analysis of variance (ANOVA) computed the regional distribution for significant group differences. 

In a second analysis step, stakeholders’ evaluation of structures on a scale from 0 to 10 was summarised to a net promotor score (NPS) [52]. The NPS uses the categorisation of promotors (rating = 9–10), passives (rating = 7–8), and detractors (rating = 0–6). The NPS is a widely recognized assessment tool in the industry, chosen for its high benchmark value among promoters. Rating passives as “adequate” or “good enough” has less positive impact and creates a wider margin between recommendation and rejection [53]. Our goal was to gain a clear understanding of who truly supports the service and who does not. The computation of the NPS can be expressed as follows:(1)$$NPS=\frac{\sum_{i}P_{i}}{n}-\frac{\sum_{j}D_{j}}{n}.$$
$\sum_{i}P_{i}$ describes the sum of all promotors, $\sum_{j}D_{j}$ specifies the sum of all detractors, and n is the total number of participants. The NPS covers the range from −100 (not at all recommended) to +100 (strongly recommended). A value of less than 0 indicates a poor rating. Values greater than 0 indicate a good rating, greater than 20 a very good rating, greater than 50 an excellent rating, and greater than 80 an outstanding rating [54].

## 3. Results

### 3.1. Sociodemographic Characteristics of the Study Population

The web-based survey was sent to 15,909 individuals, but after removing contacts due to undeliverable emails, incorrect addresses, and data deletion requests, 13,899 participants remained. The response rates varied across different professional groups, with GPs having a response rate of 6.7%, ASHIPs 61.1%, hospital medical directors 10.2%, local health departments 21.2%, MDESs 30.8%, rescue coordination centres 93.4%, outpatient nursing services 10.5%, nursing homes 10.6%, CCACs 25.6%, and hospital nursing managers 9.0%.

Respondents had an average work experience in their current position ranging from 7.27 years for ASHIPs to 18.81 years for GPs. Gender distribution varied across professions, with male majorities in GPs, ASHIPs, hospital medical directors, rescue coordination centres, and MDESs. Local health departments were evenly split, and most nursing stakeholders had female majorities (Table 1a,b).

Regional distribution showed higher participation from the west and south of Germany, reflecting their higher population densities compared to the east and north. Local health departments notably increased their workforce, with an average of 81.8% new hires compared to the existing staff. At the same time, hospital medical directors and rescue coordination centres reported smaller increases in new hires.

Over half of all participants used the German Corona-Warn-App ranging from 52.5% for outpatient nursing services to 81.8% for ASHIPs. Differences between participants and nonparticipants were observed in regional locations for outpatient nursing services, nursing homes, and CCACs and in gender distribution for GPs and rescue coordination centres.

### 3.2. Regional Outpatient and Inpatient Services

Participants assessed outpatient and inpatient COVID-19 services, with GPs, ASHIPs, and local health departments involved in outpatient care, while hospital medical directors and local health departments focused on inpatient services. Diagnostic centres received mixed feedback and COVID-19 wards in hospitals garnered strong endorsement across stakeholders (Table 2a,b).

GPs, ASHIPs, and local health departments were primarily involved in outpatient services, with GPs heavily participating in vaccination centres/teams (52.7%), video consultations (38.3%), and COVID-19 diagnostic practices (37.8%). Local health departments were extensively involved in diagnostic centres (70.9%), diagnostic teams (62.0%), and vaccination centres/teams (55.7%). ASHIPs were engaged in most outpatient services except for the Corona-Taxi. Hospital medical directors were also involved with diagnostic centres: 33.3% had direct involvement, 59.6% were aware of their regional availability, and 7.0% either were unaware or indicated no regional availability. With regard to new inpatient services, hospital medical directors (92.1%) and local health departments (40.5%) showed high involvement in COVID-19 wards in hospitals.

The rating of these services (NPS) varied significantly among stakeholders. GPs and hospital medical directors did not recommend diagnostic centres (−31.0 and −10.5) but these were positively endorsed by ASHIPs and local health departments (10.0 and 45.6). Video consultations received a high recommendation from ASHIPs (63.6) but were less favoured by GPs (−2.1), and local health departments needed to be more involved. COVID-19 diagnostic practices and COVID-19 focus practices received positive ratings across all participant groups. Certain inpatient services exhibited notable differences in endorsement levels between hospital medical directors and local health departments, such as temporary COVID-19 treatment centres (12.5 and −66.7) and temporary COVID-19 wards in rehabilitation hospitals (−20.0 and 28.5). Overall, COVID-19 wards in hospitals received strong endorsement from both groups.

### 3.3. Regional Rescue Services

Local health departments, MDESs, and rescue coordination centres showed different participation levels and NPSs highlighting varying recommendations (Table 3).

Local health departments primarily participated in the transport of COVID-19 patients via the fire department or other predefined aid organisations (27.8%), establishing COVID-19 rescue coordination centres (19.0%), and emergency medical vehicles/emergency doctor’s cars only for COVID-19 patients (16.5%). MDESs were most involved in emergency medical vehicles/emergency doctor’s cars only for COVID-19 patients (56.2%), followed by transport of COVID-19 patients via the fire department or other predefined aid organisations (21.9%) and rescue/intensive care transport helicopters with Epi-Shuttle (21.9%). Rescue coordination centres showed high participation rates in emergency medical vehicles/emergency doctor’s cars only for COVID-19 patients (68.4%), rescue/intensive care transport helicopters only for COVID-19 patients (43.9%), and rescue/intensive care transport helicopters with Epi-Shuttle (42.1%).

Establishing COVID-19 rescue coordination centres had differing NPSs among the three groups, with local health departments and MDESs showing negative scores (−22.2 and −26.7). In contrast, rescue coordination centres assigned a positive score (41.7). Similar disparities were observed for other services, such as transporting COVID-19 patients via predefined aid organisations and rescue/intensive care transport helicopters.

While no service was unanimously rejected, rescue coordination centres recommended nearly every service except tele-emergency physicians (0.0). MDESs had the highest recommendation for tele-emergency physicians (36.4) and transport of COVID-19 patients via the fire department or other predefined aid organisations (12.5) but gave negative ratings to the remaining services. Besides their top three services, local health departments were not significantly involved in others to provide substantial relevance to their NPS.

### 3.4. Regional Nursing Services

Nursing stakeholders reported an overall low level of participation in new nursing services. Despite low participation rates, services like digital appointment scheduling received positive feedback, while the majority showed varying perceptions and utilisation levels (Table 4).

Outpatient nursing services primarily engaged in crisis intervention teams (11.6%), cross-facility/cross-sector emergency nursing services or other supplementary outreach/consultation services (10.7%), and telecare services (7.4%). Nursing homes had reported higher participation rates, notably in crisis intervention teams (25.0%), digital appointment scheduling for nursing facilities (20.0%), and preventive teams (15.0%). Similarly, CCACs commonly cited crisis intervention teams (30.6%), preventive teams (14.3%), and digital appointment scheduling for nursing facilities (9.2%). Hospital nursing managers showed the highest participation in in-house nursing emergency services or other supplementary outreach/consultation services (17.2%), COVID-Team-Time-Out (17.2%), and digital appointment scheduling for nursing facilities (13.8%).

About half of local health departments regularly provided courses and consultations on hygiene measures and infection prevention for outpatient nursing services, both before (48.1%) and during (44.3%) the COVID-19 pandemic. Over half of them offered similar services for nursing homes, both before (62.0%) and during (58.2%) the pandemic.

Most surveyed services had low participation rates, such that few participants provided an NPS. Interestingly, there was a significant discrepancy in the NPS for cross-facility/cross-sector emergency nursing services, with outpatient nursing services alone giving a negative score (−7.7). In contrast, nursing homes and support centres assigned positive scores (25.0 and 50.0). Services with relatively high approval rates included digital appointment scheduling for nursing facilities, telemedicine services, preventive teams, and crisis intervention teams. Among these, crisis intervention teams received the highest participation and NPS from outpatient nursing services (33.3), nursing homes (33.3), and CCACs (20.0).

### 3.5. Communication Channels and Platforms

Communication channels varied widely among stakeholders, with telephones being the most utilised, followed by email and video calls being prominent for certain groups, and fax and on-site communication being notable, indicating diverse preferences and usage patterns (Figure 1). A detailed breakdown of the communication channels by group can be found in the appendix (Table A1, Table A2, Table A3, Table A4, Table A5, Table A6, Table A7, Table A8, Table A9 and Table A10).

The telephone was widely used, ranging from 58.0% for CCACs to 96.7% for local health departments. Email was the second-most used channel, except for GPs (32.5%) and nursing homes (35.3%), which preferred fax the second-most (GPs: 37.9% and nursing homes: 36.8%). Video calls were prominent for ASHIPs (44.6%), MDESs (33.2%), rescue coordination centres (23.6%), and hospital nursing managers (26.7%). The fax was third-most utilised by hospital medical directors (27.3%), local health departments (32.8%), and nursing services (29.1%). On-site communication was highest in nursing homes (32.8%), local health departments (25.7%), and GP practices (25.0%), while ASHIPs (11.6%), rescue coordination centres (10.3%), and CCACs (4.9%) reported the lowest. Messenger usage was generally low, with ASHIPs (8.3%), MDESs (7.6%), and hospital nursing managers (6.9%) using it more than nursing services (1.7%), nursing homes (1.3%), and CCACs (0.0%). Local health departments (45.2%), hospital medical directors (33.7%), and ASHIPs (33.0%) had the highest average usage across all channels, followed closely by hospital nursing managers (32.77%), nursing homes (32.55%), and GPs (31.9%). Communication platforms functioned primarily as one-way channels, where stakeholders and healthcare professionals received information about the evolving pandemic dynamics. The majority of participants relied on updates from the Robert Koch Institute (RKI), along with scientific publications, evidence-based guidelines, and public media. In contrast, fewer than half of the participants received informed through newsletters or updates from local health departments (Table A11 and Table A12).

## 4. Discussion

### 4.1. Key Findings

This study addresses and compares services across different German healthcare sectors during the COVID-19 pandemic. We identified 36 newly established services, from which highest healthcare stakeholder participation rates were calculated for the outpatient and inpatient sectors. Overall, services closely linked to already existing services of the pre-pandemic period consistently received better ratings. In addition, the COVID-19 pandemic may have accelerated the digitalisation of communication channels towards emails and video calls, whereby the telephone remained the most widespread communication medium.

### 4.2. Pandemic Preparedness—Transition from Evidence to Healthcare Strategies

The COVID-19 pandemic revealed gaps in the pandemic preparedness of healthcare systems requiring stakeholders to rapidly implement or adapt a pandemic plan during the ongoing pandemic [55]. A detailed retrospective analysis of pandemic-driven healthcare services is crucial for future pandemic preparedness, as a premature adoption of normality could lead to a resurgence of the outbreak [56]. This study disentangled new regional services while highlighting a notable heterogeneity regarding their levels of recommendation from a stakeholder perspective. Our findings add to the need for evidence-based strategies and ongoing consideration in translating learnings into healthcare strategies during the post-pandemic recovery phase [47] and showed that new regional services varied in their feasibility. Healthcare implementation often considers context sensitivity and consideration of clinical settings to be crucial for the acceptance and long-term adoption of new services [48,57]. The German containment strategy course highlighted the minimal consideration given to the stakeholder perspective and the potential issues that arose from this. However, the lack of integration of stakeholders in pandemic management and the lack of transparency of decisions and the relevant decision-makers was prevalent worldwide [58].

In the early phase of the pandemic, pandemic management laid emphasis on the early diagnosis of patients in order to limit the spread of the virus as effectively as possible leading to the early development of COVID-19 rapid antigen tests [59,60]. We found diagnostic centres and diagnostic practices as commonly established services while the latter were prioritized by our survey. Containment measures were extended when mRNA vaccination campaigns became more relevant [61]. As conventional healthcare sites were initially not prepared for the high volume of patients and the technical requirements of the vaccines, vaccination centres were established [6]. The majority of GPs and ASHIPs in our study participated in vaccination centres but did not recommend them. This is interesting, since it contradicts the high satisfaction levels among healthcare workers in other countries [62]. One reason could be that German GPs were bothered about disparities in remuneration, as vaccination centres were often reimbursed tenfold for a shot [63]. Also, the participants in our study preferred services that were closer to their working environment. Further outpatient services, such as specialized transport vehicles and emergency nursing services, were crucial but strained personnel resources, highlighting the challenges and importance of these measures during the pandemic [20,31,64,65]. Inpatient services showed targeted care for COVID-19 patients through restructured wards and temporary treatment centres [8,12].

The high participation and heterogeneous recommendations of our participants represent stakeholders with frontline experience, who are able to provide valuable insights and ensure that the policies reflect public values, thereby avoiding the pitfalls of late-stage engagement observed during the COVID-19 pandemic worldwide [58].

### 4.3. Traditional Communication Channels: Still Vital Amidst the Pandemic?

Our survey of communication channels indicated that personal contact could not be entirely avoided, despite the high risk of infection, possibly due to the limitations of telemedicine [66]. Interestingly, we found that digital communication channels were least utilised among GPs and the nursing sector, reflecting a contrast with other industrialised countries such as Norway and Australia [39,40]. This discrepancy may stem from Germany’s initial state of digitalisation, suggesting the need for further investment in digital infrastructure to facilitate the widespread adoption of remote consultations [38]. Such disparities in digitalisation are widespread worldwide [67,68]. Bureaucracy turned out to be another obstacle, as hospitals in the United States would have to obtain a disaster authorization before using telemedicine services, which could lead to dangerous delays in patient care [67]. Almost all participants relied on information from the RKI. However, only a minority from each group kept up to date with information from local health departments, indicating a possible gap in awareness of local changes in pandemic management.

Above all, most countries still lack a regulatory framework for telemedicine services [68]. One strategy to promote telemedicine services in daily practice is the integration of telemedicine into international and national public health guidelines, including regulations and funding, as well as operational and clinical guidelines, use cases and data sharing mechanisms for epidemiological surveillance [68]. For that purpose, telemedicine must be further developed by linking it to clinical healthcare models, further training for healthcare professionals and appropriate funding [69]. 

The permanent integration of video calls in healthcare requires comprehensive national and local strategies, including flexible funding and professional training, to harness their potential for innovative and cost-effective treatment methods [49,69,70], although some lighthouse projects in the COVID-19 pandemic were already able to prove their cost efficiency [71]. While video calls prove beneficial for specific consultations, the decreased face-to-face communication may hinder shared patient understanding and teamwork effectiveness [39,72]. However, video calls mainly saw widespread adoption across all healthcare sectors, facilitating reduced personal contact with patients in outpatient settings, decreased hospital visitors, and streamlined transportation to hospitals, notably benefiting nursing care and reducing relapses and admissions [18,23,25,35,36], even though the telephone proved to be the most important means of communication in our study. Successful implementation of digital communication still needs to overcome technical challenges, which remain a notable obstacle among healthcare professionals and patients [36,73].

### 4.4. Heterogeneity in Organisational Structures—Healthcare in a Federal System

In Germany, healthcare governance is decentralized and shared among the federal and state levels, along with corporatist self-governing bodies [74]. This heterogeneity of healthcare in a federal system can tailor services to differences in regional demographic and health care structures. The current German pandemic plan by the RKI—a subordinate agency of the Federal Ministry of Health responsible for the surveillance, detection, prevention and control of diseases—refers to the federal structure of states and their responsibilities for pandemic surveillance, control and management [50]. The RKI’s current pandemic plan provides planning aids for hospitals, rescue services and nursing homes with regard to staff protection, hygiene, organization and area-specific measures. For the outpatient sector, reference is made to the responsibility of the ASHIPs resulting in 17 different pandemic plans. This multiplicity of plans increases bureaucracy and yields inconsistencies, where people face different regulations when moving between different states. 

A major challenge in the national health goals process is the binding implementation of these goals. Recent legislative measures in Germany have provided important impetus by recognizing national health goals as a reference for defining action areas and criteria for primary prevention and health promotion services, indicating a step towards more cohesive implementation across all levels [75]. The US National COVID-19 Preparedness Plan exemplifies a comprehensive strategy that includes enhancing vaccine distribution, establishing “One-Stop Test to Treat” locations, updating preventive measures, strengthening data analysis, protecting vulnerable populations, and promoting healthcare equity [76].

### 4.5. Strengths and Limitations

This study offers a comprehensive overview of COVID-19 response services across healthcare sectors, with a broad representation of professionals through a consistent survey and a substantial sample size. For that purpose, a questionnaire was developed to measure suitable and valid information in this pandemic situation across important healthcare sectors in Germany. The Net Promoter Score (NPS), originally derived from market research, has been successfully integrated into a scholarly context here. A high response rate among the rescue coordination centres was presumably achieved by distributing the invitation to tender via the newsletter of the federal association of rescue coordination centres. A low response rate from GPs, who also started the vaccination campaign at the time of the survey, and nursing professionals limits the results, although the total number of participants was still large.

## 5. Conclusions

During the COVID-19 pandemic, Germany experienced significant variation in the implementation of pandemic-related services across healthcare sectors, with stakeholders prioritising services built on existing healthcare structures. Outpatient and inpatient services showed the highest implementation and participation rates among healthcare professionals. While telephone communication was most commonly used, the pandemic necessitated a reactive surge in telemedicine and digital communication due to infection dynamics. Developing a proactive digital infrastructure connecting healthcare professionals across sectors is crucial for improved pandemic management. Integrating pandemic-related and telemedical services into national and international public health guidelines, ensuring their feasibility and cost-efficiency, and overcoming technical challenges, particularly in digital communication, are essential steps forward. This study focused on new services in the German healthcare system, but many of these structures have already been implemented in other countries, suggesting that the main findings may also be applicable in an international context. The research highlights the need for further studies to assess the long-term sustainability and impact of pandemic-related services, particularly in diverse healthcare systems, and to explore how these innovations can be integrated as a backup beyond emergency situations. Future research should prioritize evaluating past COVID-19 measures from the perspectives of stakeholders to inform improvements in national and regional pandemic preparedness.

## Figures and Tables

**Figure 1 healthcare-12-02192-f001:**
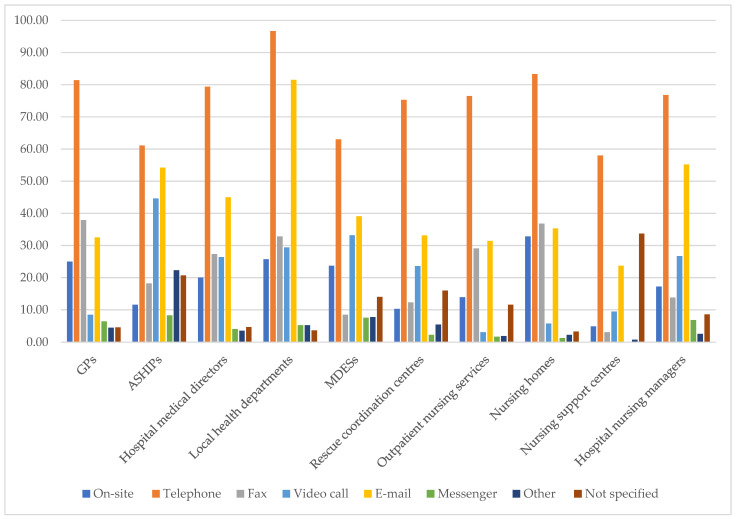
Average use of communication channels by common groups in the healthcare system.

**Table 1 healthcare-12-02192-t001:** (a) Sociodemographic and workplace characteristics. (b) Sociodemographic and workplace characteristics.

(a)					
	GPs N = 630	ASHIPs N = 11	Hospital Medical Directors N = 114	Local Health Departments N = 79	MDESs N = 73
Personal					
Sex male/female/diverse, n (%)	364 (57.8)/263 (41.7)/ 3 (0.5)	8 (72.7)/3 (27.3)/ 0 (0.0)	105 (92.1)/9 (7.9)/ 0 (0.0)	38 (48.1)/41 (51.9)/ 0 (0.0)	64 (87.7)/9 (12.3)/ 0 (0.0)
Years in current position, mean (SD)	18.81 (9.61)	7.27 (4.29)	11.06 (6.60)	11.82 (10.43)	8.86 (5.77)
Region					
North, n (%)	104 (16.5)	2 (18.2)	16 (14.0)	14 (17.7)	10 (13.7)
South, n (%)	173 (27.5)	3 (27.3)	30 (26.3)	22 (27.8)	17 (23.3)
East, n (%)	127 (20.2)	2 (18.2)	27 (23.7)	17 (21.5)	12 (16.4)
West, n (%)	226 (35.9)	4 (36.4)	41 (36.0)	26 (32.9)	34 (46.6)
Workplace					
Number of employees, mean (SD)	8.02 (8.83)	-	1052.58 (1358.61)	62.18 (58.21)	-
New hires due to COVID-19, mean (SD)	-	-	15.95 (49.56)	50.88 (85.21)	-
**(b)**					
	**Rescue Coordination Centres** **N = 57**	**Outpatient Nursing Services** **N = 121**	**Nursing Homes** **N = 100**	**CCACs** **N = 98**	**Hospital Nursing Managers** **N = 29**
Personal					
Sex male/female/diverse, n (%)	57 (100.0)/0 (0)/ 0 (0.0)	27 (22.3)/94 (77.7)/ 0 (0.0)	36 (36.0)/63 (63.0)/ 1 (1.0)	11 (11.3)/86 (87.8)/ 1 (0.9)	9 (31.0)/20 (69.0)/ 0 (0.0)
Years in current position, mean (SD)	9.53 (7.07)	13.27 (9.18)	9.53 (8.15)	8.81 (6.03)	8.62 (7.22)
Region					
North, n (%)	9 (16.1)	30 (24.8)	15 (15.0)	24 (24.5)	3 (10.3)
South, n (%)	25 (44.6)	12 (9.9)	33 (33.0)	23 (23.5)	9 (31.0)
East, n (%)	8 (14.3)	34 (28.1)	20 (20.0)	11 (11.2)	5 (17.2)
West, n (%)	14 (25.0)	45 (37.2)	32 (32.0)	40 (40.8)	12 (41.4)
Workplace					
Number of employees, mean (SD)	46.77 (26.05)	33.21 (32.61)	64.44 (45.04)	3.80 (3.07)	528.93 (617.18)
New hires due to COVID-19, mean (SD)	0.65 (2.60)	-	-	-	-

**Table 2 healthcare-12-02192-t002:** (a) Outpatient services by group—involvement and rating. (b) Inpatient services by group—involvement and rating.

(a)												
	GPs (N = 630)				ASHIPs (N = 11)				Local Health Departments (N = 79)			
	A (%)	B (%)	D (%)	NPS	A (%)	B (%)	D (%)	NPS	A (%)	B (%)	D (%)	NPS
Diagnostic centres	23.0	66.7	10.3	−31.0	90.9	9.1	0.0	10.0	70.9	24.1	5.1	45.6
Diagnostic teams	14.4	43.2	42.4	−22.8	63.6	9.1	27.3	−14.3	62.0	10.1	27.8	22.5
COVID-19 diagnostic practices	37.8	39.5	22.7	8.7	72.7	18.2	9.1	50.0	21.5	48.1	30.4	58.8
Corona-Taxi	5.4	28.3	66.3	17.6	45.5	0.0	54.5	0.0	6.3	12.7	81.0	40.0
COVID-19 outpatient clinics	15.6	36.2	48.3	−8.2	81.8	0.0	18.2	0.0	19.0	30.4	50.6	26.6
COVID-19 focus practices	16.3	38.3	45.4	43.7	81.8	9.1	9.1	33.4	13.9	40.5	45.6	36.3
Video consultations	38.3	43.5	18.3	−2.1	100	0.0	0.0	63.6	1.3	32.9	65.8	−100.0
Vaccination centres/teams	52.7	45.6	1.7	−13.3	63.6	27.3	9.1	−28.5	55.7	41.8	2.5	54.6
Other (outpatient)	23.5	14.9	61.6	25.7	25.0	25.0	50.0	−100.0	21.2	18.2	60.6	14.3
**(b)**												
	**Hospital medical directors (N = 114)**				**Local Health Departments (N = 79)**							
	**A (%)**	**B (%)**	**D (%)**	**NPS**	**A (%)**	**B (%)**	**D (%)**	**NPS**				
COVID-19 wards in hospitals	92.1	7.0	0.9	47.6	40.5	58.2	1.3	84.4				
COVID-19 focus hospitals	34.2	28.9	36.8	17.9	12.7	27.8	59.5	40.0				
COVID-19 wards in hospitals stratified by care level	53.5	17.5	28.9	32.7	12.7	50.6	36.7	70.0				
Temporary COVID-19 treatment centres	7.0	15.8	77.2	12.5	3.8	2.5	93.7	−66.7				
Temporary COVID-19 wards in rehabilitation hospitals	8.8	29.8	61.4	−20.0	17.7	17.7	64.6	28.5				
Other (inpatient)	12.8	12.8	74.4	55.6	3.9	5.9	90.2	100.0				

A = Regionally present, involved; B = Regionally present, not involved; D = Not existent or known.

**Table 3 healthcare-12-02192-t003:** Rescue services by group—involvement and rating.

	Local Health Departments (N = 79)				MDESs (N = 73)			Rescue Coordination Centres (N = 57)		
	A (%)	B (%)	D (%)	NPS	C (%)	D (%)	NPS	C (%)	D (%)	NPS
Establishing COVID-19 rescue coordination centres	19.0	12.7	68.4	−22.2	20.5	79.5	−26.7	21.1	78.9	41.7
Emergency medical vehicles/emergency doctor’s cars only for COVID-19 patients	16.5	41.8	41.8	9.1	56.2	43.8	−4.9	68.4	31.6	5.1
Transport of COVID-19 patients only via the fire department or other predefined aid organizations	27.8	15.2	57.0	36.4	21.9	78.1	12.5	22.8	77.2	23.1
Tele-emergency physicians	3.8	10.1	86.1	0.0	15.1	84.9	36.4	10.5	89.5	0.0
Rescue/intensive care transport helicopters only for COVID-19 patients	1.3	3.8	94.9	−33.4	15.1	84.9	−27.2	43.9	56.1	8.0
Rescue/intensive care transport helicopters with Epi-Shuttle	1.3	1.3	98.7	0.0	21.9	78.1	−31.2	42.1	57.9	37.5
Dedicated medical response teams for COVID-19 patients	-	-	-	-	2.7	97.3	−66.7	-	-	-
Other types of patient transfer across healthcare sectors	2.0	0.0	98.0	100.0	20.3	79.7	7.7	25.5	74.5	8.3
New organization of transfer management	0.0	4.2	95.8	-	19.0	81.0	63.6	17.6	82.4	22.3
New form of communication and cooperation across healthcare sectors	0.0	12.5	87.5	-	37.1	62.9	28.6	22.0	78.0	40.0

A = Regionally present, involved; B = Regionally present, not involved; C = Regionally present without indication of involvement; D = Not existent or known.

**Table 4 healthcare-12-02192-t004:** Nursing services by group—involvement and rating.

	Outpatient Nursing Services (N = 121)			Nursing Homes (N = 100)			CCACs (N = 98)			Hospital Nursing Managers (N = 29)		
	C (%)	D (%)	NPS	C (%)	D (%)	NPS	C (%)	D (%)	NPS	C (%)	D (%)	NPS
Digital appointment scheduling to visit nursing facilities	-	-	-	20.0	80.0	26.3	9.2	90.8	0.0	13.8	86.2	75.0
Telenursing services	7.4	92.6	0.0	5.0	95.0	−40.0	1.0	99.0	0.0	3.4	96.6	0.0
Teletherapy services	0.0	100.0	-	3.0	97.0	66.7	2.0	98.0	100.0	0.0	100.0	-
Telemedicine services	1.7	98.3	0.0	13.0	87.0	7.7	4.1	95.9	25.0	10.3	89.7	66.7
Other new digitally supported nursing services	5.8	94.2	85.7	12.0	88.0	66.7	12.2	87.8	58.3	10.3	89.7	100.0
Preventive teams	-	-	-	15.0	85.0	13.3	14.3	85.7	30.7	-	-	-
Crisis intervention teams	11.6	88.4	33.3	25.0	75.0	33.3	30.6	69.4	20.0	-	-	-
Cross-facility/cross-sector emergency nursing services or other supplementary outreach/consultation services	10.7	89.3	−7.7	8.0	92.0	25.0	9.2	90.8	50.0	3.4	96.6	0.0
COVID-Team-Time-Out	-	-	-	-	-	-	-	-	-	17.2	82.8	100.0
In-house nursing emergency services or other supplementary outreach/consultation services	-	-	-	-	-	-	-	-	-	17.2	82.8	40.0
Other (nursing)	1.7	98.3	100.0	3.0	97.0	66.7	6.1	93.9	16.7	0.0	100.0	-

C = Regionally present without indication of involvement; D = Not existent or known.

## Data Availability

The data used and analysed are available on reasonable request from the corresponding author.

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
