# Peer review of "Innovative Regional Services and Heterogeneous Communication Channels: Results from the Nationwide German egePan Project for Pandemic Management"

_healthcare, 2024, doi:10.3390/healthcare12212192_

Round 1

Reviewer 1 Report

Comments and Suggestions for Authors

This report represents a nationwide sample of ten German healthcare stakeholders regarding their use of the novel COVID-19 regional services and communication channels that emerged across all sectors. There was significant variation in implementation with an internationally comparable low level of digitalization of healthcare and a high level of telephone communication. Developing proactive digital infrastructure is the recommendation. The suggestion is that future research uses this type of evaluation provided in the report to enhance national and regional pandemic preparedness.

The work is well-conceived, well-written, well-analyzed, well-referenced, and thoughtfully interpreted. Conceptually, the weaknesses are minor. 

Line by line suggested edits

57 As citation 29 is to a 2010 reference, the authors must either delete citation 29 or, if they consider this reference seminal, they should indicate in the text why they are citing a 2010 reference regarding COVID-19.

68 Germany is considered a highly developed nation. Therefore, the authors should explain why Germany differed from Norway and Australia in embracing digital solutions and telemedicine.

88 Please state in what way this research is novel.

123 Please describe the criteria used to develop this web-based questionnaire.

124 Please cite the most relevant research from the comprehensive literature review that influenced the development of the web-based questionnaire. 

130 Please state the method for obtaining the email addresses.

187 Please provide additional information about selecting the NPS for this study and include a current reference.

200 The authors are asked to explain the low return of GPs in the text.

202 The authors are asked to explain the high return of rescue coordination centers in the text.

341-342 This statement addresses how the research is novel. Please move it to line 88. Furthermore, the authors must reference the other “few” studies and indicate how this research differs.

393-399 A reason is provided here for the request in line 68 for Germany’s backwardness regarding digital communication. Please mention these points comparing Germany’s digitalization with other countries in the Introduction regarding line 68. 

451 The information about GPs starting their vacation is the type to answer the request in line 200 and should be provided at that point.

501-684 Please refer to the Instructions for Authors and redo the references according to the required style. Please italicize the titles, bold the year of publication, and italicize the journal number:  https://www.mdpi.com/journal/healthcare/instructions.

Reviewer 2 Report

Comments and Suggestions for Authors

Engagement in addressing new threats, including infectious diseases, is crucial for ensuring an effective healthcare response. Measuring this engagement, along with how it is perceived by stakeholders such as healthcare providers and medical staff, is essential for improving future preparedness. The use of tools like the Net Promoter Score (NPS) in healthcare and health sciences is a positive development, as it provides valuable insights into stakeholder satisfaction and the effectiveness of new services. Applying such metrics helps to identify strengths and areas for improvement, enhancing the overall response to emerging health threats.

The introduction is well-prepared and enriched with up-to-date literature. The aim of the study is clearly stated at the end of the introduction.

The methodology is described clearly, comprehensively, and correctly. However, it would be beneficial to clarify the types of questions in the questionnaire and the response options. There is detailed information about the specialists involved in developing the questionnaire, but there is a lack of specific details regarding the questionnaire itself as well as the inclusion and exclusion criteria.

The results are presented appropriately. The tables are clear and easy to read, as is the graph. They are well-described in the text.

The discussion is properly prepared. The study's limitations are mentioned only in relation to the small sample size in two groups—general practitioners and nursing staff—though I believe the outcome is still significant.

The conclusions are too brief and require a more comprehensive connection to the results.

Reviewer 3 Report

Comments and Suggestions for Authors

The manuscript entitled “Innovative regional services and heterogeneous communication channels: Results from the nationwide German egePan project for pandemic management” was interesting. The authors aimed to describe newly established services in response to the COVID-19 pandemic from a stakeholder perspective and to examine the interprofessional communication channels applying a nationwide cross-sectional approach. The following comments can help the authors to improve it:

1-      The objective of the study should be clear in the abstract and introduction section, preferably with the same wording.

2-      Please choose the keywords based on the MeSH terms.

3-      A number of new diagnostic centres in non-medical settings have been mentioned in the introduction. Were these centres implemented in Germany? If not, what were the newly implemented services in Germany?

4-      The number of eligible participants in each group should be mentioned in the methods section.

5-      More information about the questionnaire development is required. What were the used references? What about the reliability and validity assessment of the questionnaire? In addition, please provide a copy of the questionnaire as an Appendix instead of explaining it in section 2.3 and 2.4.

6-      Instead of the term “Gender”, please use “sex” in table 1a, and add related information for Female participants.

7-      What are the research implications?

8-      How can other countries use the results for making their future strategies?

9-      Regarding the research innovation, I think the items of the questionnaire and the results are at a very basic level and can’t convey important messages to the readers.

Round 2

Reviewer 3 Report

Comments and Suggestions for Authors

I appreciate the authors for their time and efforts to revise the manuscript. However, as a reader, I am still concerned about the adequacy of data analysis, the application of the results, and the inovative side of the research.
